# Advanced Methods for Kiln-Shell Monitoring to Optimize the Waelz Process for Zinc Recycling

Markus Vogelbacher [1,*,†] , Sina Keller [2,†] , Wolfgang Zehm [3,†] and Jörg Matthes [1,†]

1 Institute for Automation and Applied Informatics, Karlsruhe Institute of Technology (KIT), Hermann-von-Helmholtz-Platz 1, 76344 Eggenstein-Leopoldshafen, Germany; joerg.matthes@kit.edu
2 ci-Tec GmbH, Beuthener Str. 16, 76139 Karlsruhe, Germany; s.keller@ci-tec.de
3 Befesa Zinc Freiberg GmbH, Alfred-Lange-Strße 10, 09599 Freiberg, Germany; wolfgang.zehm@befesa.com
* Correspondence: markus.vogelbacher@kit.edu; Tel.: +49-721-608-25750
† These authors contributed equally to this work.

**Abstract:** The recycling of zinc in the Waelz process is an important part of the efficient use of resources in the steel processing cycle. The pyro-metallurgical processing of zinc-containing wastes takes place in a Waelz rotary kiln. Various measured variables are available to monitor the process. The temperature of the kiln-shell is analyzed by an infrared kiln-shell-scanner. In this paper, methods are presented which eliminate external weather-related disturbances on the temperature measured by the kiln-shell-scanner using a weather model and which extend the monitoring of the regularly necessary melting process to remove accretions. For this purpose, an adapted sigmoid estimation is introduced for the melting process, which provides new information about the current process status and a forecast of the further development of the melting process. As an assistance system for the plant operator, this enables an efficient execution of the melting process and reduces downtimes.

**Keywords:** process supervision; zinc recycling; kiln-shell; weather model; temperature model; process monitoring; energy saving

## 1. Introduction

A large proportion of the zinc produced worldwide is applied as corrosion protection for steel. Since the use of galvanized steel, for example, in the automotive industry, will continue in the future, the return of steel scrap will not stop. Accordingly, the recycling of steel scrap and the extraction of zinc are of great importance. In this context, a cycle has developed that enables resources to be used efficiently. When recycling steel scrap in the steelworks, zinc-containing flue dust is generated. The furnace's high temperatures mean that the zinc evaporates due to its low boiling temperature and reoxidizes in the exhaust gas flow. In the Waelz process [1], these dusty steel mill residues are converted into so called Waelz oxides which represent a zinc-concentrate with a high amount of zinc oxide.

An essential part of the Waelz process is the pyro-metallurgical processing of the zinc-containing wastes in the Waelz rotary kiln [2]. Through a slight incline and approximately one rotation per minute of the rotary kiln, which is up to 65 m long, the feeding mixture is conveyed through the kiln with a residence time of 4 h to 6 h. The mixture is dried and heated by the hot kiln gas, which is fed in counterflow. In the central kiln area in the reaction zone, the zinc, lead, and iron, are reduced at temperatures of around 1100 °C to 1200 °C [3]. While the iron remains in the slag and is reoxidized at the end of the rotary kiln, the zinc is evaporated into the headspace. This evaporation is caused by the high temperatures and vapor pressures and reoxidized with the oxygen in the kiln atmosphere. The raw gas, which is rich in metal oxides, is finally subjected to exhaust gas treatment, cooled, and the Waelz oxide is separated on a filter [4].

Variations in the composition of the feeding mixture can significantly affect the entire process. They lead, for example, to accretions on the kiln brickwork and thus complicate

the process control. Various process variables are available for the plant operator to support the process control. Such process variables are, for example, the speed of the fan of the air supply or the throughput. In addition, camera-based parameters are used. A particular infrared camera system [5], which provides information about the current slag temperature [6], is available to control the air supply at the end of the kiln [7]. This supply significantly influences the temperature of the kiln gas. The formation of accretions is observed using a kiln-shell-scanner. An infrared-line-scanner measures the temperature of the kiln-shell from the outside [8]. The entire kiln-shell is displayed via rotation, enabling the detection of hot spots or rings. Previous work has mainly used information about the kiln-shell-scanner to detect local effects [9]. Yang and Sun [10] use a variant of the Hilbert transformation on the kiln-shell-scanner data in order to predict the point in time and the location of a refractory failure. In [11], the rotary kiln's brickwork of a cement plant is protected from overheating with the help of the kiln-shell-scanner. The local temperature distribution is measured and a target temperature is maintained by water cooling. Sadighi et al. [12] and Muhammed and Al-Yasiri [13] use the kiln-shell-scanner temperature to estimate the accretion thickness inside the rotary kiln considering the process parameters and the outside/inside temperature heat transfer. Based on the thickness estimation, the authors in [14] detect local defects on the wall of the rotary kiln. The previous work uses the temperatures of the kiln-shell-scanner without considering the influence of external weather-related disturbances on the kiln-shell temperature measurement, which is part of this paper.

The kiln-shell-scanner data are crucial for the melting process, which is necessary several times a year. The energy-intensive process has not yet been part of any work in the literature. The progress of the melting process is monitored by the kiln-shell-scanner and is heavily dependent on human judgement and the expert knowledge of the plant operator. The human assessment often leads to the melting process being performed longer than necessary in the individual zones, which results in higher energy consumption and unnecessarily more extended downtime. For the Waelz rotary kiln systems in zinc recycling, no studies on further possible uses of the kiln-shell-scanner data beyond monitoring exist.

This paper examines the extended application possibilities of the kiln-shell-scanner data due to the importance of the kiln-shell-scanner data for accretion monitoring, especially during melting processes. To ensure that the transmitted data is correct, the influence of external weather-related disturbances on the kiln-shell-scanner temperature is first analyzed, and a model for disturbance suppression is set up. The previously new possibility of analytical evaluation and documentation of the melting process is considered with the corrected kiln-shell-scanner data. For this purpose, a temperature model is set up that enables a forecast of the development of the melting process and can thus support the plant operator as an assistance system.

## 2. Melting Process for Reduction of Accretions

The Waelz process is negatively influenced by large grown accretions. Once they get too thick, the accretions are treated by melting them down. The progress of the melting process is monitored by observing the temperature profile along the rotary kiln using a kiln-shell-scanner. The first designs of the kiln-shell-scanner have been equipped with pyrometers, and hot spots in the rotary kiln are detected by displaying peak temperatures [15]. With further developments, infrared-line-scanners for monitoring and analyzing kiln brickwork have been established [8,16,17].

The plant operator controls the melting process based on experience, supported by the temperature profile of the kiln-shell-scanner. During an optimal melting process, the accretions on the kiln brickwork are systematically removed from kiln feeding to the end. Such a melting process begins with emptying the kiln and introducing coke. First, a large amount of air is introduced into the kiln at high negative pressure, and the oxygen is fed into the rear and middle kiln area. As a result, the coke that builds up behind the accretions starts to burn more intensely. The increase of the temperature in the accretions ensures

that the contained metallic iron melts, and thus the accretions break down. Depending on kiln-shell temperature development, the negative pressure can be gradually reduced, and the melting process can be controlled from kiln feeding to the end. Finally, accretions at the end of the kiln can be melted and broken down using an additional burner at the kiln outlet. The accretion free rotary kiln is the ideal starting point for the Waelz process.

Depending on the process situation, a melting process has to be executed several times a year. Each time, a high amount of resources, especially coke, is necessary. The end of the melting process is difficult to estimate for the plant operator. In practice, it is melted until no changes in the temperature profile of the rotary kiln can be recognized. It is currently not taken into account that hot material still in the rotary kiln has a positive effect on the melting progress. As a result, the melting process could be ended earlier and the remaining accretions could be melted down by the residual material. The consideration of this effect and the presentation of an appropriate information is part of this paper.

## 3. Data Basis for Kiln-Shell Monitoring at Zinc Recycling

In addition to melting processes, this paper analyzes the influence of weather on the shell temperature of the Waelz rotary kiln and thus on the kiln-shell-scanner temperature. For this reason, measurement data under constant process conditions are necessary, which can be tracked using the data from the process control system. These include, for example, the speeds of various fans to influence the process air and the negative pressure, the slag temperature, or the rotation speeds of the kiln power unit. For constant process conditions, there should be no kiln downtime and an almost constant slag temperature.

The data basis for this paper are measurements of a kiln-shell-scanner PYROLINE 256 L/256 Hz from DIAS Infrared GmbH [18] on a rotary kiln for zinc recycling from August 2019 and February to March 2020. The temperatures are stored in a matrix that is created by unwinding the temperature values from the rotary cylinder (Figure 1).

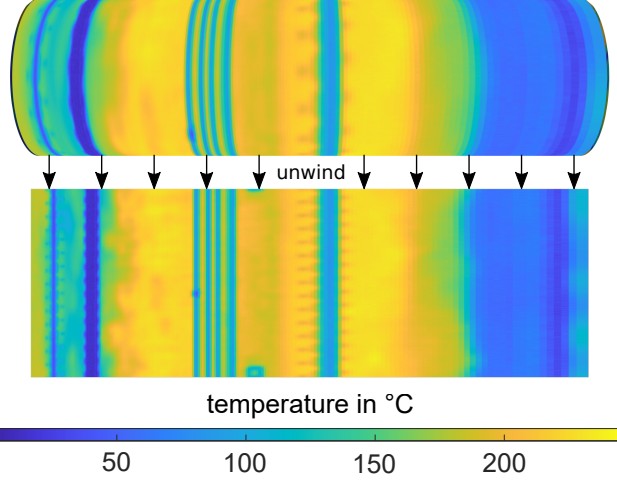

**Figure 1.** Unwind kiln-shell to a two-dimensional kiln-shell temperature matrix.

For process documentation, the temperature matrices are saved every hour. The selected time periods contain on the one hand constant process conditions over several days and on the other hand melting processes. For further processing, the temperature matrix is divided into individual temperature zones along the longitudinal axis of the rotary kiln. Figure 2 shows the partition in different temperature zones. At date $t$ and in zone $z$, the mean kiln-shell temperature is defined as $KS_T(t, z)$.

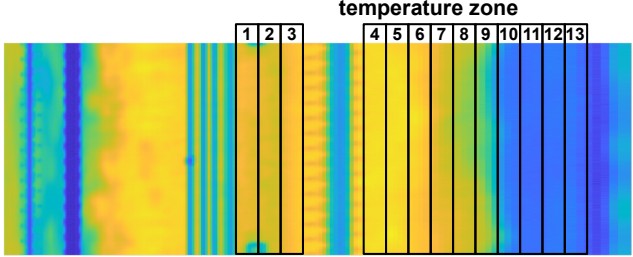

**Figure 2.** Visualization of the selected kiln-shell temperature zones.

Since there are no accretions in the first third of the rotary kiln (inlet), this area is not considered for the temperature zones. In addition, the cold bearings of the rotary kiln are omitted.

For the periods considered, the weather conditions on the Waelz rotary kiln are given by the data from a nearby weather station (www.timeanddate.de, accessed on 10 April 2021). The temperature is listed in degrees Celsius and the humidity in percent at intervals of half an hour. The closest weather value is used for the comparison with the kiln-shell-scanner data.

Table 1 gives an overview of the data used for evaluation in the measurement periods from August 2019 and February to March 2020.

**Table 1.** Used data basis for analyzing zinc recycling processes.

| Measurement System | Measured Variables | Availability |
|---|---|---|
| process control system | process air<br>negative pressure<br>slag temperature<br>rotation speed kiln | every minute |
| kiln-shell-scanner | shell temperature | every hour |
| weather station | outside air temperature<br>air humidity | every half hour |

## 4. Advanced Analysis of Kiln-Shell Data

The kiln-shell-scanner data represent an essential source of information for the plant operator to evaluate the zink recycling process. In a first step, described in Section 4.1, data disturbances need to be identified and eliminated. Subsequently, in Section 4.2, the corrected data can then be used for extended analysis, such as the melting process. Figure 3 shows the flow diagram of the data processing presented in this paper.

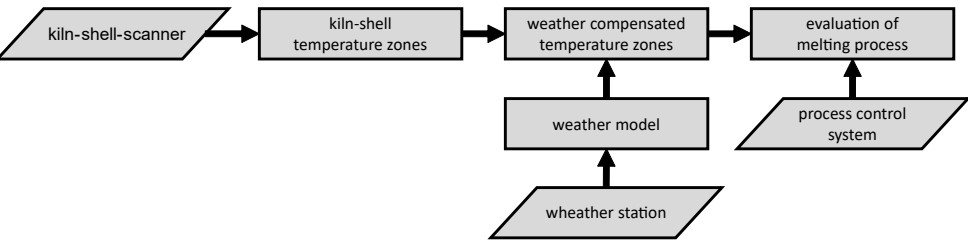

**Figure 3.** Flow diagram of data processing.

### 4.1. Eliminating Data Disturbances—Weather Compensation

In order to identify data disturbances, the zero-mean temperature profile of the defined temperature zones is examined for time periods with almost constant process conditions (Figure 4a). For example, the slag temperatures vary steadily between 1000 °C to 1200 °C and the rotation speeds of the kiln power unit between 400 rpm to 600 rpm, as shown in Figure 4b.

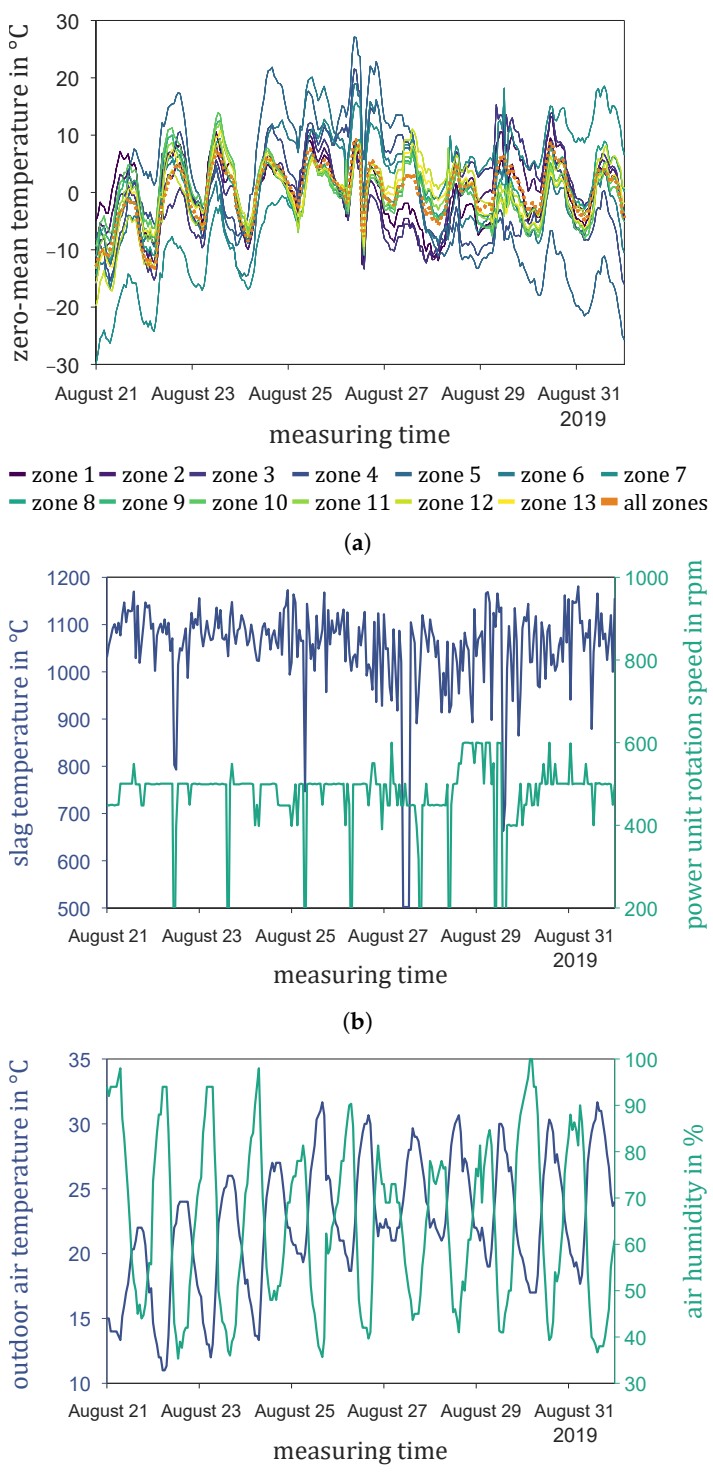

**Figure 4.** Visualization of (**a**) the zero-mean temperatures of different kiln-shell zones shown in Figure 2; (**b**) the process data of slag temperature and kiln power unit rotation speed; and (**c**) the weather data containing the outdoor air temperature and air humidity.

The representation of the temperatures from all zones in Figure 4a illustrates the similarity of the temperature profiles. Differences only occur in the middle temperature zones (zones 4–6), since higher temperatures and less accretions are to be expected in these areas due to the process. Figure 4c shows the outdoor air temperature and the air humidity for the considered time period. Both variables show a wave-like course, with the maxima shifted by half a day.

Despite constant process conditions, a similar characteristic applies for the progress of the zero-mean temperature of different zones and mean temperature of the entire rotary kiln-shell for all zones in Figure 4a. When comparing the zero-mean temperature and the outdoor air temperature, shown in Figure 4c, a linkage between the kiln-shell-scanner temperature and external weather-related influences are present. Local maxima of the zero-mean temperature profile occur mainly at noon, while the local minima are at night. These occurrences follow the diurnal variation of the air temperatures.

In addition to the apparent similarity of the temperature profiles (see Figure 4a,c), we consider the correlation coefficients between the mean kiln-shell-scanner temperature of all zones and the outdoor air temperature. A correlation coefficients is characterized by values in the range between $-1$ to $1$. The value $0$ refers to no correlation, $1$ corresponds to a high positive, while $-1$ to high negative correlation. The correlation coefficient between the mean kiln-shell scanner temperature and the outdoor air temperature results in a value of $0.88$, implying a positive correlation and the coefficient between the mean kiln-shell scanner temperature and the air humidity a value of $-0.75$ implying a negative correlation.

Based on these findings, a kiln-shell temperature model $KS_T^c(t,z)$ with weather compensation can be set up. This model contains the measured kiln-shell temperature $KS_T(t,z)$ corrected by a weather-influenced temperature part $W(t,z)$:

$$KS_T^c(t,z) = KS_T(t,z) - W(t,z), \tag{1}$$

with a date $t$ and a temperature zone $z$ of the rotary kiln.

The linear weather model $W(t,z)$ includes the influence of the outdoor air temperature $O_T(t)$ and the air humidity $H(t)$ on a date $t$:

$$W(t) = a_1 O_T(t) + a_2 H(t). \tag{2}$$

According to Figure 4a, all temperature zones are characterized by similar profiles, which indicates that they are equally influenced by the weather conditions. Thus, the weather-influenced temperature part of the model can be written as $W(t)$. Since the the diurnal variation is mainly caused by weather influences, the zero-mean temperatures are used to estimate the parameters of the weather model $W(t)$. Furthermore, to reduce the local effects within individual zones, e.g., the breaking of accretions, the mean temperature of the entire rotary kiln-shell $KS_T(t)$ is used. The parameter estimation is performed using a least-squares estimation for 10 days in August with constant process conditions, as shown in Figure 4b.

Figure 5a visualizes the comparison of weather model $W(t)$ and zero-mean measured data $KS_T(t)$ for the 10 days in August selected for parameter estimation and Figure 5b shows the comparison between the measured kiln-shell temperatures $KS_T(t)$ and the kiln-shell temperatures corrected by the weather-related part $KS_T(t) - W(t)$.

The alternating part caused by weather influences can be removed from the kiln-shell temperatures and thus provide better information about the current development of the temperature profile.

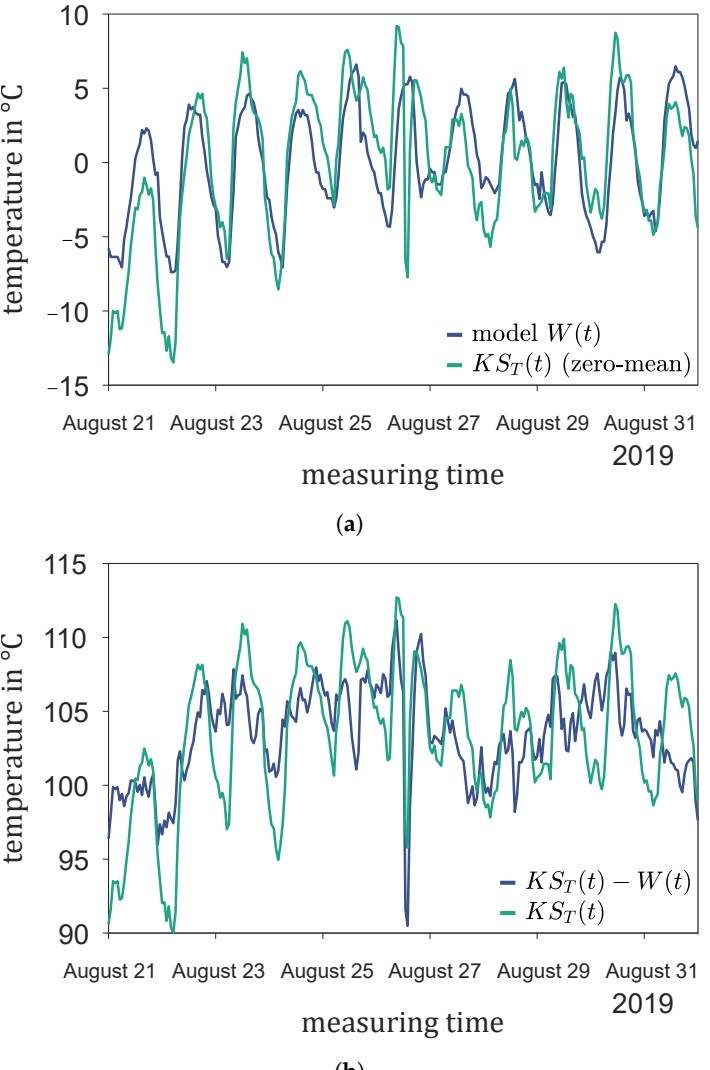

**Figure 5.** Visualization of (**a**) the weather model data compared to the measured data of the kiln-shell temperature (zero-mean); (**b**) the measured data of the kiln-shell temperatures compared to the kiln-shell temperatures corrected by the weather-related part.

### 4.2. Evaluation of the Melting Process of Accretions

Monitoring the kiln-shell temperatures is crucial for the analysis of a melting process. Figure 6 shows a melting process for the different weather-compensated temperature zones $KS_T^c(t, z)$.

The reduction of accretions from the beginning of the rotary kiln to the end becomes visible through the different points in time of the temperature rise for the individual zones. For the analytical consideration of the melting process, the profile of the temperatures can be converted into a model using an adapted sigmoid function $KS_{T,sig}$:

$$KS_{T,sig} = \frac{b_1}{1 + e^{(-b_2(t - b_3))}} + b_4. \tag{3}$$

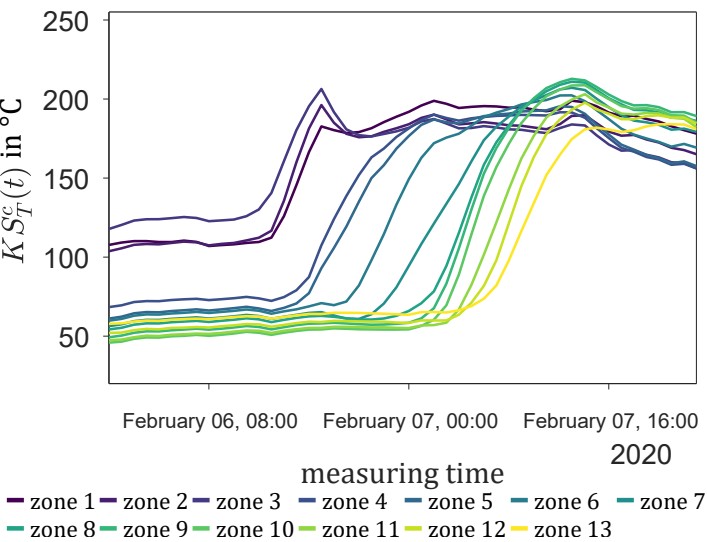

**Figure 6.** Temperature zones' profiles of a melting process in February 2020.

An example is shown in Figure 7.

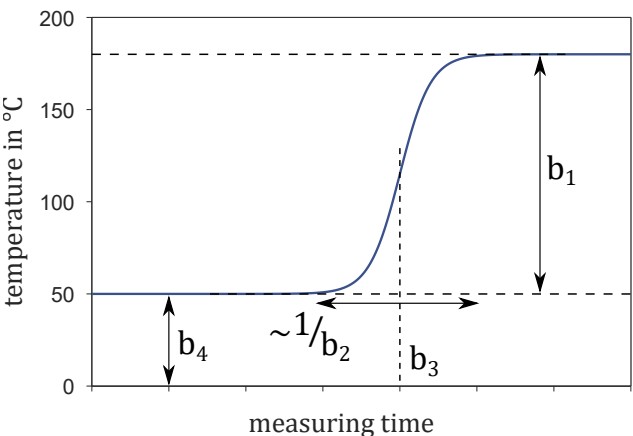

**Figure 7.** Example of an adapted sigmoid function as a model for the temperature of a zone during a melting process.

The parameters $b_1 \ldots b_4$ can be read directly from the sigmoid function (Figure 7) and interpreted as physical quantities of the melting process. $b_1$ describes the temperature rise, $b_2$ the duration, $b_3$ the inflection point, and $b_4$ the temperature at the beginning of the melting process. For the identification of the parameters $b_1 \ldots b_4$ based on current measurement data, it makes sense to define reasonable parameter ranges and initial values. The ranges and initial values can be determined to be empirically based on historic data and expert process knowledge. For the temperature rise $b_1$, there is a range between 40 and 250 °C, for the inflection point $b_3$, the time range between the beginning of the melting process and 72 h later, and, for the start temperature $b_4$, a range between 30 and 150 °C is given. The range of values for parameter $b_2$ is defined between 0.2 and 3 and is inversely proportional to the melting duration. $b_4$ has the current temperature at the beginning of the melting process as an initial value. The remaining initial values can be selected from the specified ranges.

By nonlinear least-squares estimation of $b_1 \ldots b_4$, the determination of the sigmoid function can be carried out at any point in time during the melting process. Figure 8 shows such an intermediate result for the previously shown melting process. The predicted temperature profiles (red line) from the estimated sigmoid functions give a good approximation of the future true profile (dashed line).

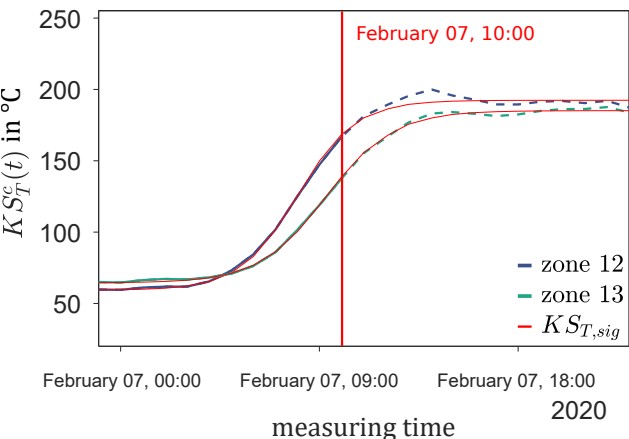

**Figure 8.** Results of the sigmoid estimation (red line) for a point in time during the melting process compared to the current and future temperature profile (dashed line).

Thus, information about the future profile of the melting process can be obtained on the basis of the estimated model. The derivation of the sigmoid function (Figure 9, green line)

$$\frac{\mathrm{d}KS_{T,sig}}{\mathrm{d}t} = \frac{b_1 b_2 e^{(-b_2(t-b_3))}}{(1 + e^{(-b_2(t-b_3))})^2} \tag{4}$$

shows how far the melting process has progressed and even the probable end can be predicted. For this, empirical conditions for the end are specified. On the one hand, a previously defined temperature rise $\Delta\vartheta$ must have been reached compared to the start of the melting process. On the other hand, the inflection point of the sigmoid function has to be exceeded and a minimal derivation $\varepsilon$ (Figure 9, red line) has to be reached:

$$t > \arg\max_t \frac{\mathrm{d}KS_{T,sig}}{\mathrm{d}t}, \quad \frac{\mathrm{d}KS_{T,sig}}{\mathrm{d}t} \leq \varepsilon. \tag{5}$$

$\varepsilon$ is determined empirically and can be adapted to the specific plant by expert knowledge. In addition to the end, the beginning of the melting process can also be estimated using the derivation. When the derivation reaches the minimal value $\varepsilon$ before the inflection point for the first time, this is defined as the beginning (Figure 9). Together with the end, the total duration of the melting process can be specified.

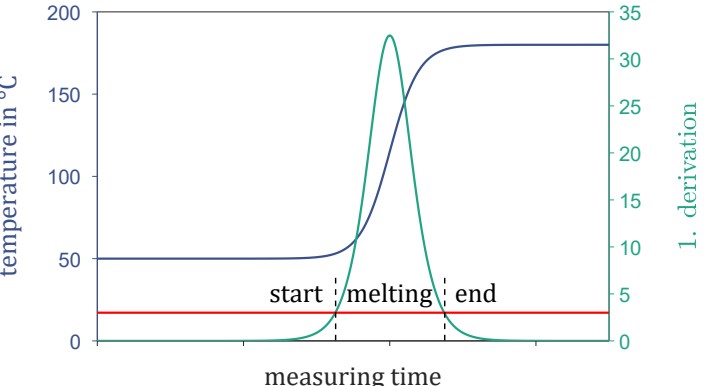

**Figure 9.** 1. Derivation (**green**) of a sigmoid function (**blue**) with minimal limit (**red**) for the estimated start and end of the melting process.

If the hot coke leaves the first zones, temperature drops can occur, as can be seen in Figure 6. Since this cannot be covered by the adapted sigmoid model, the temperature profile for the last two hours is analyzed in detail as an additional secondary condition. If the temperature profile decreases monotonically for the defined time range before the

conditions in Equation (5) are reached, the melting process is also considered to have ended. $KS_T^c(t - 120\text{min}, z)$, $KS_T^c(t - 60\text{min}, z)$ and $KS_T^c(t, z)$ are used in this article for the analysis for temperature data available every hour.

The sigmoid estimation of a zone is trusted as soon as the parameters $b_1 \ldots b_4$ converge to a nearly constant value within the limits $\pm\Delta b_i$ for $i = 1 \ldots 4$. Figure 10 shows an example of the temporal profile of the parameter $b_1$ for the melting process from Figure 6. During the melting process, the parameters of the estimated sigmoid functions reach a constant value and thus a reliable sigmoid estimation.

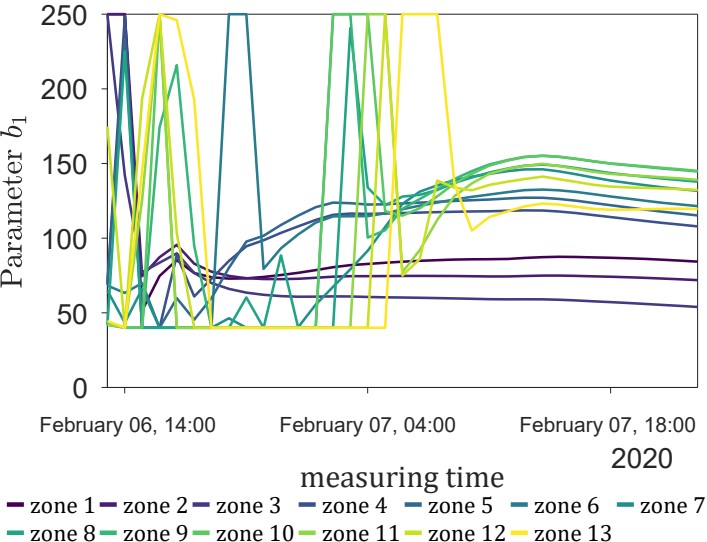

**Figure 10.** Temporal profile of parameter $b_1$ for sigmoid estimation of a melting process in February 2020.

If the parameters are within the limits for a period of e.g., two hours, the sigmoid estimation is stable, and the entire melting process can be analyzed using the sigmoid estimation. The end time of the melting process can thus be reliably determined.

## 5. Results and Discussion

Previous work (see Section 1) used the data from the kiln-shell-scanner directly. The influence of weather-related disturbances is not examined. The automatic evaluation of the melting process in the zinc area is also not part of previous work. For this reason, no comparison with existing work is possible. Practical examples are used to evaluate the process and compared with the experience of experts about the process. Previous work can benefit from the weather-related disturbance suppression presented in this paper and optimize its results.

To evaluate the weather model, the temperature profile is considered after a melting process. It is expected that accretions will slowly build up again and thus the kiln-shell-scanner temperature will decrease again over time. Figure 11a shows the raw data of the mean kiln-shell-scanner temperature (green line) for such a profile after the melting process and the associated weather model (blue line) for this period. The raw data contain significant increases in temperature, which cannot be explained by the process data, but are also visible in the weather model. In contrast, for the corrected weather-compensated temperatures in Figure 11b, no increases can be recognized. The temperature is monotonically falling as expected because of the continuous build up of accretions. It turns out that the influence of external weather-related disturbances can be eliminated by the weather model and thus it provides better information about the current state of the accretions.

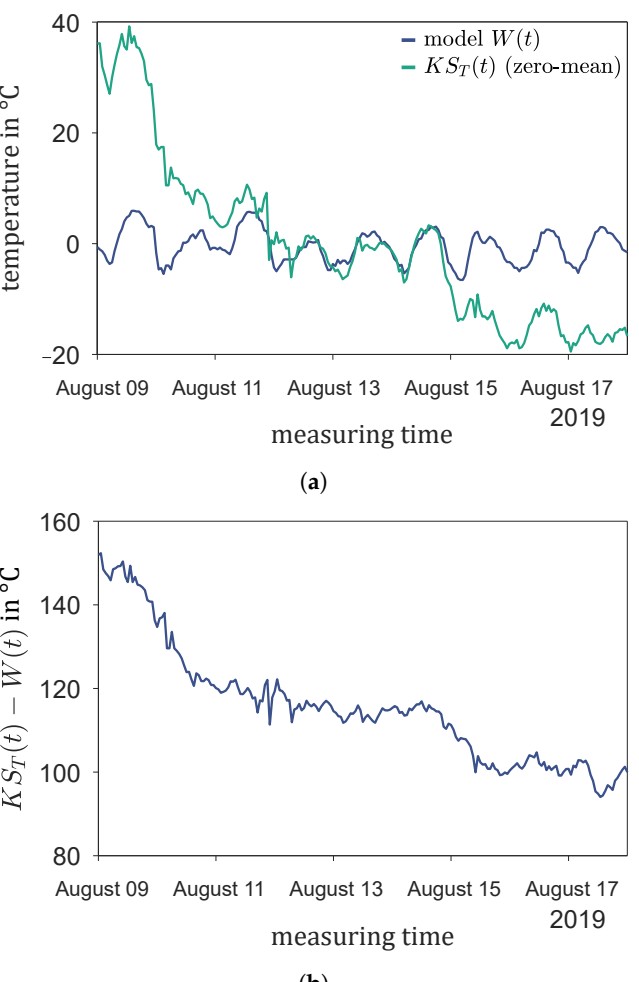

(a)

(b)

**Figure 11.** (**a**) Comparison of raw data of the mean kiln-shell-scanner temperature and the associated weather model; (**b**) weather-compensated mean kiln-shell-scanner temperature.

An analysis of kiln-shell temperature according to Section 4.2 is executed for the melting process in February 2020 (Figure 6) and for another one on the same kiln in March 2020 (Figure 12).

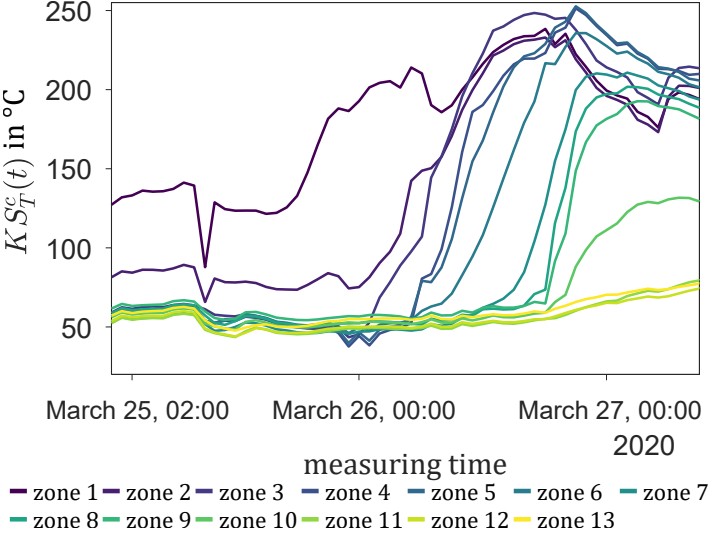

**Figure 12.** Profile of the temperature zones of a melting process in March 2020.

At first, the profile of the derivation at each date can be specified for the sigmoid function estimated at the same date (Figure 13).

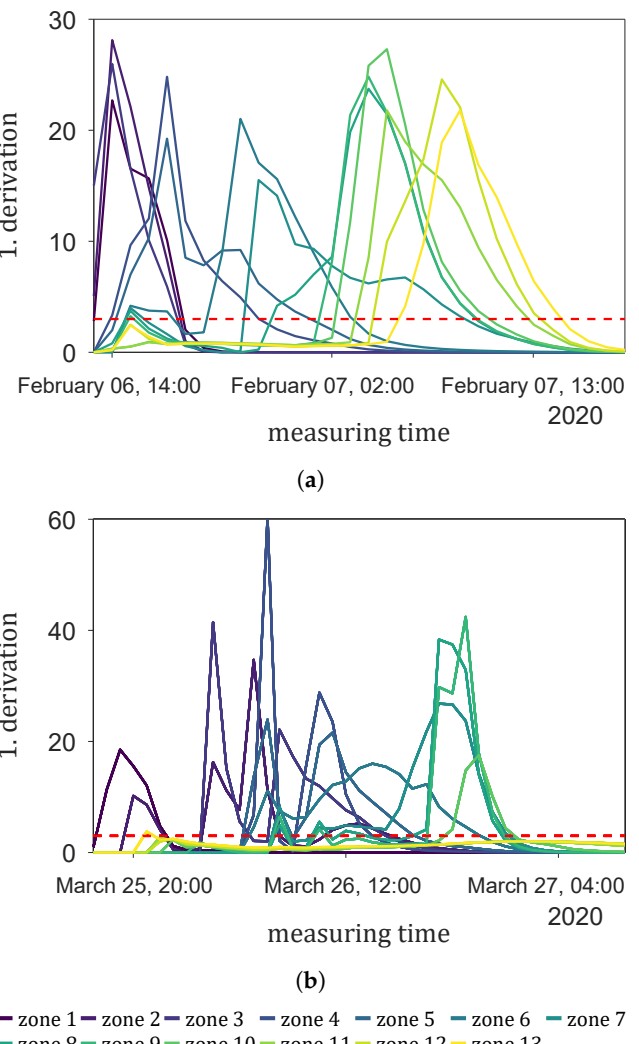

**Figure 13.** 1. Derivation of the up-to-date estimated sigmoid function for the melting processes from (**a**) Figure 6 and (**b**) Figure 12.

The sampled results take the expected profile from Figure 9 and thus enable an evaluation of how far the melting process has progressed in the individual zones. If a temperature zone has exceeded the maximum of the derivation, the temperature rise will gradually decrease over the next few hours, and a progression of the melting process into the next zone would make sense. Information about a necessary change in negative pressure at an early stage can be shown to the plant operator, and an overview can be given of whether the accretions are reduced as desired from kiln feeding to the end. If the derivation falls below the specified minimum value $\varepsilon$ (Figure 9, dotted line), the end for the corresponding zone can be documented. During the melting process, the probable end can also be predicted via the sigmoid estimation and used as information for the further process.

For the melting processes in February and March 2020, the optimal end point of the melting process could be correctly and automatically recognized for all zones. For three zones in February and four zones in March, it was even possible to predict the end point between one to three hours in advance. Figure 14 shows the profile of the temperature for these zones with the associated times of stable sigmoid estimation (red diamond) and the calculated end of the melting process from this date (blue star). The end of the melting

process is a forecast for the selected temperature zones; for the remaining zones, not shown, the end of the melting process corresponds to the point in time of stable sigmoid estimation.

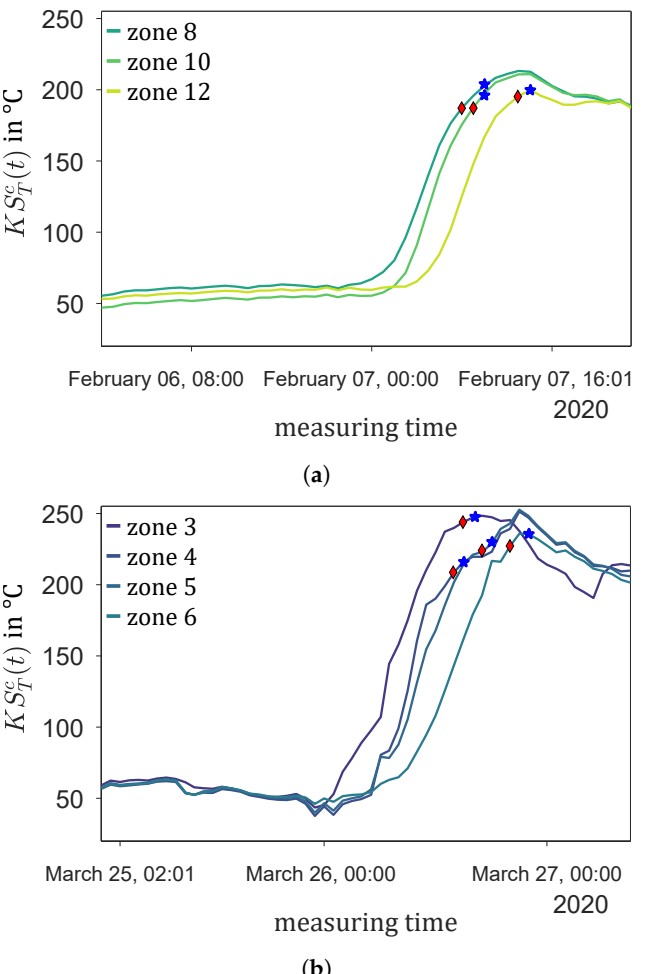

**Figure 14.** Profile of selected temperature zones for the melting process in (**a**) February and (**b**) March 2020 with labels for the time of stable sigmoid estimation (red diamond) and the calculated end of the melting process (blue star).

The estimated end is influenced by the minimum derivation value $\varepsilon$ and can be adjusted to the system-specific requirements by the plant operator.

The extended processing of the kiln-shell-scanner data by dividing the rotary kiln into different temperature zones and the estimation of adapted sigmoid functions as a model for the melting process can automatically identify or even forecast the optimal end point of the melting process, thus supporting the plant operator as an assistance system and can be used for documentation.

## 6. Conclusions

The kiln-shell-scanner temperature is an important source of information for the plant operator for evaluating accretions, for example, during a melting process. The assessment of the current process situation in normal operation, and especially during melting processes, is based on human judgement and the expert knowledge of the plant operator. A correct and error-free presentation of the current kiln-shell temperature is therefore an important factor. In this paper, a linear weather model consisting of outside temperature and air humidity is introduced that can eliminate the disturbance of external weather-related influences and thus enable a correct representation of the temperature profile. The corrected temperatures ensure a more reliable source of information for a human observation and for a subsequent automatic evaluation.

The corrected kiln-shell-scanner data can be divided into temperature zones and visualized. These temperature zones can be used to ensure the optimal sequence of a melting process, i.e., systematically removing accretions on the kiln brickwork from kiln feeding to the end. An extended monitoring is presented for melting processes that allows the current status to be assessed on the basis of a sigmoid model estimations and thus supports the plant operator. The sigmoid model contains parameters that can be interpreted as physical quantities of the melting process. The progress and end of the melting process can be identified or even forecasted automatically. The human assessment of the current status of the melting process can thereby be supported. In particular, too long lasting melting processes can be shortened and important resources, such as coke, can be saved. According to a manager of a zinc recycling plant, by reducing coke and $CO_2$ because of reducing the absolute melting time, costs of around 400 € per hour can be saved at current resource prices. Furthermore, cost-intensive downtimes can be reduced and production can be restarted earlier.

**Author Contributions:** M.V., J.M., W.Z. and S.K. contributed to the conceptualization and methodology of this research. J.M., W.Z. and M.V. have acquired the image data resources. M.V. designed the software, curated the data, performed the investigation, formal analysis, validation, and visualization of the results. S.K. and J.M. initialized the related research and provided didactic and methodological inputs. All authors prepared and cross-reviewed section-wise the original draft. All authors have read and agreed to the published version of the manuscript.

**Funding:** This research work is part of the OPTIMER project (Grant No.: 03ET1490A), which is funded by the German Federal Ministry of Economic Affairs and Energy (BMWi).

**Institutional Review Board Statement:** Not applicable.

**Informed Consent Statement:** Not applicable.

**Data Availability Statement:** Not applicable.

**Acknowledgments:** We acknowledge support by the KIT-Publication Fund of the Karlsruhe Institute of Technology.

**Conflicts of Interest:** The authors declare no conflict of interest.

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
