# Peer review of "Advanced Methods for Kiln-Shell Monitoring to Optimize the Waelz Process for Zinc Recycling"

_processes, doi:10.3390/pr9061062_

Round 1

Reviewer 1 Report

Advanced Methods for Kiln Shell Monitoring to Optimize the Waelz Process for Zinc Recycling

Correct monitoring  of the temperature on a kiln shell is essential for efficient operation of the kiln. Systems of such monitoring have been used for over 40 years and they are constantly developed along with the development of measuring equipment and computer software. The systems mainly represent the state-of-the-art in kiln shell infrared scanning. The systems consists of three main components: Infrared scanner head, data processing unit, and the software for the user.

The only unquestionable novelty brought by this article is the inclusion of local weather conditions in the rotary kiln operation analysis, and the sigmoid estimation method for analyzing the process of melting the accretions on the lining. The rest are known guesses and already used solutions (offering high reliability, and long-term stability and accuracy). The progress and end of the melting process can be identified or even  forecasted automatically, it is an interesting fulfillment of the solutions used so far and leads to the improvement of the rotary kiln monitoring.

  1.  

33 caused by the high temperatures and vapor pressures and reoxidized with the oxygen in

34 the kiln atmosphere ??The raw gas, which is rich in metal oxides, is finally subjected to

Reoxidized with the oxygen  what about  CO2, fuel is burned and there must be carbon dioxide  Zn(g)+ CO2=ZnO + CO

End of sentence “.”    between atmosphere  and The???

2

206 shell-scanner temperature will decrease again over time. Figure 11a shows the raw

207 data of the mean kiln-shell-scanner temperature (blue line) for such a profile after the

 208 melting process and the associated weather model (red line) for this period.

 No red color in the drawing, probably there was a frequent case when the drawing from Excel was saved in Word.

Author Response

Dear Reviewer,

Thank you very much for your feedback on our work. We really appreciate your review of our contribution and your helpful comments and suggestions. Your remarks were taken into account and fixed.

Reviewer 2 Report

The authors present an interesting work, however, the quality of the article is very poor and must be greatly improved for its publication in the journal. Some indications are:
• The line number is on the wrong side, have you used the magazine Template?
• The contributions of the authors must be indicated at the end of the manuscript, as indicated in the Template, the symbol placed in the names suggests that the authors have died
• The introduction is very poor, it should be expanded with more recent research, keep in mind that only 5 references appear in all of it.
• Section 2.1 cannot be called state of the art, being an explanation of only 7 lines.
• Melting Process should be described in more depth
• Line 109, abbreviations and equations must be explained
• Line 116, write degrees Celsius and percent
• Table 1, eliminate ellipsis or clarify them
• The quality of Figure 3 is very poor
• The graphs presented in the results are very poor. The units in the axes must go between parentheses, the axes must be black and use legends to differentiate the curves, the x-axis can be numeric and indicate that it is the time in days, the figures such as 4 (a) with as many graphs nothing is appreciated, the legend is not in format, the discussion of the graphs in turn is very reduced, figure 10 appears cut off ...
• It would be good to support the discussion with other similar research works that support the research
• Section 6, Conclusions, should be expanded, furthermore, where does the € 400 saving come from?
• References should be expanded, and the DOI of the articles should also be included.

Round 2

Reviewer 2 Report

The authors have made all the proposed changes, congratulations